# Enantioselective synthesis of [4]helicenes by organocatalyzed intermolecular C-H amination

Xihong Liu [1] ✉, Boyan Zhu[1], Xiaoyong Zhang[2], Hanwen Zhu[1], Jingying Zhang[1], Anqi Chu[1], Fujun Wang[1] & Rui Wang [1] ✉

Catalytic asymmetric synthesis of helically chiral molecules has remained an outstanding challenge and witnessed fairly limited progress in the past decades. Current methods to construct such compounds almost entirely rely on catalytic enantiocontrolled fused-ring system extension. Herein, we report a direct terminal *peri*-functionalization strategy, which allows for efficient assembling of 1,12-disubstituted [4]carbohelicenes via an organocatalyzed enantioselective amination reaction of 2-hydroxybenzo[*c*]phenanthrene derivates with diazodicarboxamides. The key feature of this approach is that the stereochemical information of the catalyst could be transferred into not only the helix sense but also the remote C-N axial chirality of the products, thus enabling the synthesis of [4]- and [5]helicenes with both structural diversity and stereochemical complexity in good efficiency and excellent enantiocontrol. Besides, the large-scale preparations and representative transformations of the helical products further demonstrate the practicality of this protocol. Moreover, DFT calculations reveal that both the hydrogen bonds and the C-H⋯π interactions between the substrates and catalyst contribute to the ideal stereochemical control.

As screw-shaped compounds formally derived from *ortho*-annulated aromatic and/or hetero-aromatic rings, helically chiral molecules have fascinated synthetic chemists for more than 100 years owing to their esthetic architectures and unique chiroptical properties[1–6]. Tremendous efforts have been devoted to developing the methodologies for the synthesis of enantiomerically pure helicenes, which have been demonstrated in the past decades to be highly promising for application in diverse fields, such as asymmetric catalysis[7–9], molecular recognition[10–12], molecular machines[13,14], material sciences[15–17], and some biologically active agents[18,19]. Incipiently, substrate-controlled diastereoselective cyclisation reactions were employed to induce the screw sense of the helicenes[20–27]. However, the utilizing of chiral auxiliaries or tethers inevitably led to tedious and complicated preparation of enantiomerically enriched starting materials and high

economic cost. In this regard, catalytic asymmetric protocols with high levels of stereocontrol are particularly alluring, but have proven to be extremely challenging.

While approaches to optically pure point-chiral[28] and axially chiral compounds[29,30] have mushroomed enormously in the past decades, catalytic enantioselective synthesis of helicenes[31,32] is still in its infancy, and corresponding literatures reported to date have been fairly limited (Fig. 1a). Progress in this field mainly comes from transition-metal-catalyzed [2 + 2 + 2] cycloadditions[33–42] and intramolecular hydro-arylation of alkynes[43–51]. Complementarily, sporadic transition-metal-catalyzed other approaches[52–54], such as V-catalyzed oxidative coupling of polycyclic phenols and Rh-catalyzed enantioselective C-H activation/annulation process of 1-aryl isoquinoline derivatives and alkynes were reported by Sasai and You, respectively. Besides, the rare

[1]Key Laboratory of Preclinical Study for New Drugs of Gansu Province, School of Basic Medical Sciences & Research Unit of Peptide Science, Chinese Academy of Medical Sciences, 2019RU066, Lanzhou University, 730000 Lanzhou, China. [2]Institute of Systems and Physical Biology, Shenzhen Bay Laboratory, 518107 Shenzhen, China. ✉e-mail: liuxihong@lzu.edu.cn; wangrui@lzu.edu.cn

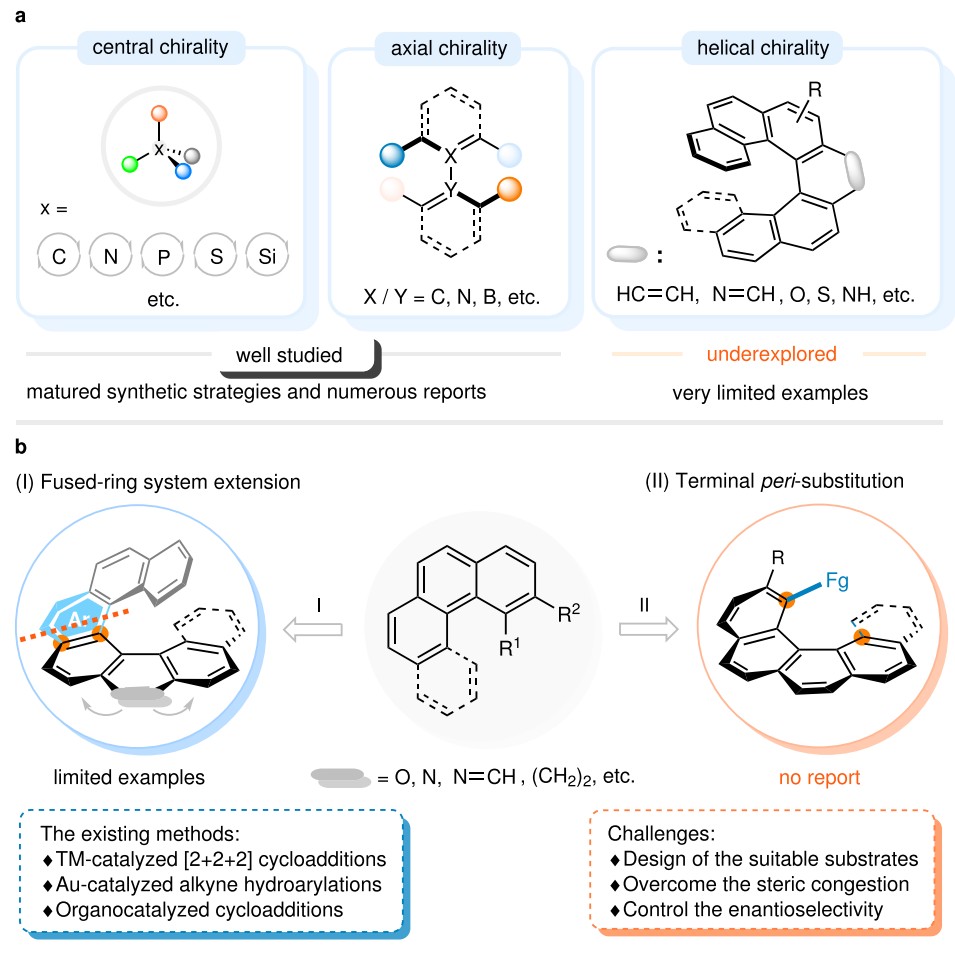

**Fig. 1 | Background introduction and our strategy for synthesizing optically pure helicenes. a** The research status of catalytic enantioselective synthesis of chemicals with different types of chiralities. **b** Strategies for the enantioselective synthesis of hetero- and carbohelicenes. TM transition metal.

examples of organocatalyst-mediated cycloadditions were also documented to be efficient to access enantiomerically enriched helicenes[55–58]. Especially, it is noteworthy that during the preparation of this manuscript, Baudoin and co-workers reported a Pd-catalyzed enantioselective intramolecular C-H arylation for access to lower carbo[n]helicenes[59], groups of Yang and Li independently published the chiral phosphoric acid catalyzed enantioselective Povarov reaction/oxidative aromatization process to allow for the synthesis of azahelicenes[60,61]. However, despite these pioneering and elegant works, nearly all the approaches mentioned above utilized a strategy of extending fused-ring systems through stereoselective cyclizations to construct configurationally stable helicenes (Fig. 1b−I)[33–61]. The scarcity of complementary assembly strategies has significantly hindered the on-demand synthesis of helicenes or helicene-like molecules through catalytic enantioselective methods. Therefore, the exploration of more innovative and practical concepts for obtaining optically pure helicenes with a wide structural diversity is highly desired.

It should be noted that besides π-conjugated scaffold extension through arene formation, terminal *peri*-functionalization of the choreographed substrates could also generate configurationally stable [4]- and [5]helicenes. However, such a tactic has hitherto not been applied in the catalytic enantioselective synthesis of functionalized helicenes and their heteroanalogues (Fig. 1b-II), which might arise from three reasons: (1) there is few or no appropriate and available substrate; (2) the large steric hindrance caused by terminal ring makes it difficult to install a substituent group at the fjord region of the helical precursors;

(3) unlike building stereogenic centers, the control of helical chirality is more challenging because it is a phenomenon of nanoscale, which calls for the pursuit or design of rational catalyst to obtain satisfactory enantioselectivity. To realize this vision, benzo[c]phenanthren-2-ol **1** with a substituent at its 12-position, in which the hydroxyl group could function as both a directing group and a binding site, was designed and synthesized (Fig. 2). We speculated that diverse functional groups could be incorporated at the 1-position in a catalytic helicoselective fashion by utilizing the nucleophilicity of the enol tautomer of substrate **1**, thereby increasing the barrier to enantiomerization and giving access to enantioenriched 1,12-disubstituted [4]helicenes. If such an endeavor meets with success, a series of highly functionalized helicene molecules that had previously been unexploited or inaccessible would be easily assembled via an intermolecular electrophilic aromatic substitution reaction.

For this purpose, we speculated that highly reactive azo-compound[62] might be a suitable electrophile to realize the functionalization of highly steric-hindered fjord-type area of polycyclic phenol **1**. To the best of our knowledge, catalytic asymmetric amination reaction of phenols with azo-compounds represents one of flexible and versatile protocols for preparing functionalized molecules with different stereogenic elements. For example, groups of Jørgensen and Tan independently realized the catalytic asymmetric synthesis of C-N axially chiral compounds by employing a direct C-N axis construction strategy and a desymmetrisation strategy, respectively (Fig. 2a-I)[63–66]. Besides, You group, Luan group, and Pan group successively reported

**a.** Catalytic enantioselective amination reactions between β-naphthols and azo-compounds

construction of C-N stereogenic axes

construction of chiral quaternary carbon centers

**b.** Catalytic enantioselective amination of polycyclic phenols: efficient synthesis of helicenes (this work)

- organocatalysis
- terminal *peri*-substitution
- high enantioselectivities
- broad substrate scope

**Fig. 2 | Diverse synthesis of compounds with different chiralities via catalytic enantioselective amination reactions of polycyclic phenols. a** Previous works: construction of axially and centrally chiral compounds by Friedel-Crafts and dearomative aminations. **b** This work: construction of helically chiral molecules by developing a terminal *peri*-amination strategy. CPA chiral phosphoric acid.

the catalytic asymmetric assembling of aza-quaternary carbon centers by utilizing dearomative amination reactions of α-substituted β-naphthols with either azodicarboxylates or diazodicarboxamides (Fig. 2a-II)[67–69]. However, despite these conspicuous progresses, the applications of similar chemical processes in helicenes synthesis have not yet been reported, even in a non-helicoselective version. To further extend the potential of Friedel-Crafts amination reactions in asymmetric synthesis and enrich the synthetic strategies of enantioenriched helicenes, we reported herein an enantioselective terminal *peri*-amination strategy for the efficient synthesis of configurationally stable [4]- and [5]helicenes (Fig. 2b).

## Results

### Optimization of reaction conditions

Initially, the amination reaction of 12-methylbenzo[*c*]phenanthren-2-ol **1a** and diazodicarboxamide **2a** was selected as model reaction to examine the feasibility of our concept. Considering the extraordinary performance of bifunctional organocatalysts in asymmetric catalysis[70,71], we envisage that a double-activation mode via H-bonding interactions could be applied to promote the depicted pathway and control the stereochemistry. Accordingly, typical Takemoto thiourea catalyst **C1** was first employed to drive the current reaction, and pleasingly, furnished the desired [4]helicene **3a** in 41% yield with 67.5:32.5 er at −50 °C (Table 1, entry 1). As expected, the C-H amination of **1a** indeed brought a sufficient increase in the energy barrier of racemization, leading to the formation of a pair of enantiomers that were configurationally stable enough and not interconvertible at room temperature. Motivated by this result, commercially available cinchona alkaloid-derived thiourea **C2** and squaramide **C3**, as well as (*S*,*S*) −1,2-cyclohexanediamine-derived thioureas **C4** and **C5** were evaluated

to further improve the enantioselectivity (entries 2-5). However, only **C4** delivered a slightly increased er value. In the light of relatively larger size of helical topology, it was conjectured that extending the steric hindrance group of the catalyst might avail to the long-range control of enantioselectivity. With this in mind, catalyst **C6** bearing a 1-pyrenyl substituent was synthesized and provided a significantly improved enantioselectivity (entry 6, 86:14 er). The subsequent solvent screening revealed a preference for THF (entries 7–9). When the concentration and scale of the reaction mixtures were increased, the same excellent outcome was obtained (entry 11, 45% yield, 96.5:3.5 er).

### Substrate scope

Having established the optimal reaction conditions, we sought to investigate the substrate scope and limitations of this intermolecular amination reaction. A variety of diazodicarboxamides **2** were examined in combination with 12-methylbenzo[*c*]phenanthren-2-ol **1a** (Fig. 3). Either electron-donating or electron-withdrawing groups at the *para*-position of the phenyl ring were all compatible with this protocol, affording the corresponding 1,12-disubstituted [4]helicenes in high yields and enantioselectivities (**3b**-**3j**, 37-46% yield, 95.5:4.5-98:2 er). Changing the position of methyl substituent on the phenyl ring had no apparent effect on the reaction results (**3k**-**3l**). Pleasingly, the reactions of diazodicarboxamides with a disubstituted phenyl ring occurred in satisfactory yields and enantioselectivities (**3m**-**3p**). Only in the case of **3n** was a relatively diminished yield (28%) observed, probably resulting from rapid decomposition of substrate **2n**. Moreover, diazodicarboxamide bearing a bulkier naphthalene group also worked well (**3q**). Importantly, the substituent of substrate **2** has not to be aromatic in nature; alkyl-substituted substrates, such as those with benzyl, *n*-hexyl and cyclohexyl, were all smoothly converted to the

**Table 1 | Optimization of the reaction conditions[a]**

| Entry | Catalyst | Solvent | t (h) | Yield[b] (%) | er[c] |
|---|---|---|---|---|---|
| 1 | C1 | toluene | 2 | 41 | 67.5:32.5 |
| 2 | C2 | toluene | 2 | 41 | 61:39 |
| 3 | C3 | toluene | 2 | 45 | 54.5:45.5 |
| 4 | C4 | toluene | 3 | 47 | 70:30 |
| 5 | C5 | toluene | 2 | 45 | 39:61 |
| 6 | C6 | toluene | 4 | 45 | 86:14 |
| 7 | C6 | CH₂Cl₂ | 6 | 43 | 78.5:21.5 |
| 8 | C6 | THF | 6 | 45 | 96.5:3.5 |
| 9 | C6 | Et₂O | 24 | 41 | 90.5:9.5 |
| 10[d] | C6 | THF | 5 | 45 | 96.5:3.5 |
| 11[e] | C6 | THF | 6 | 45 | 96.5:3.5 |

[a]Reaction conditions: **1a** (0.1 mmol, 1.0 equiv), **2a** (0.05 mmol, 0.5 equiv), and catalyst (5 mol%) in 1.0 mL of solvent at −50 °C. THF: tetrahydrofuran.
[b]Isolated yield based on **1a**.
[c]The er value of **3a** was determined by chiral HPLC analysis.
[d]Carried out in 0.5 mL of THF.
[e]Carried out on 0.2 mmol scale in 1.0 mL of THF.

desired enantioenriched [4]helicenes with excellent outcomes (**3r-3t**, 38–44% yield, 96:4-98:2 er), which further highlighted the compatibility of this transformation.

Next, we targeted the exploration of substrate scope with respect to 2-hydroxybenzo[*c*]phenanthrenes **1** (Fig. 4). The effect of substituents at position 12 was first examined. Besides methyl, ethyl and

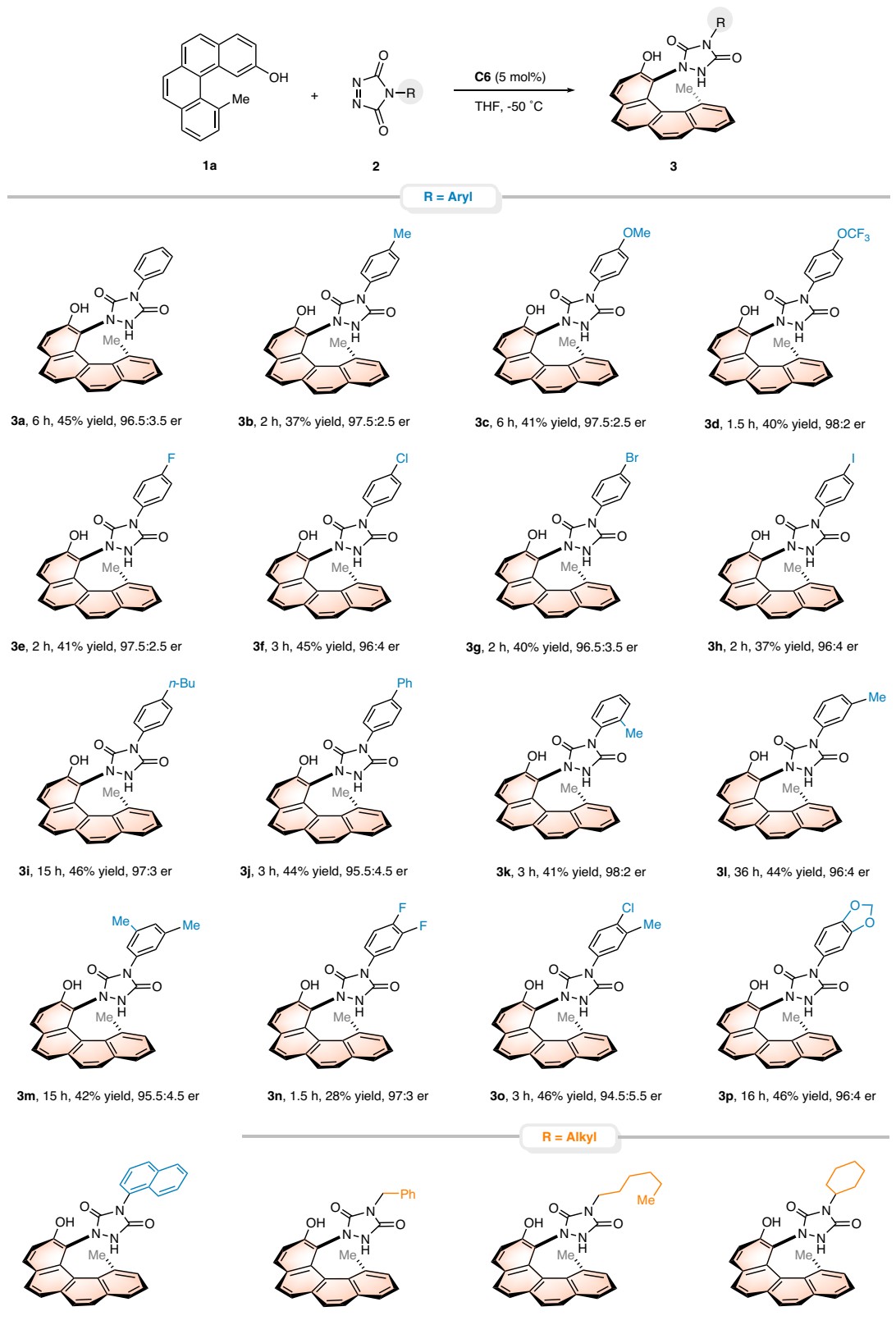

**Fig. 3 | Substrate scope with respect to diazodicarboxamides.** Reaction conditions: 0.2 mmol **1a** (1.0 equiv), 0.1 mmol **2** (0.5 equiv), and 5 mol% **C6** in THF (1.0 mL) at −50 °C. Isolated yields based on **1a** are shown. The er values were determined by HPLC analysis using a chiral stationary phase.

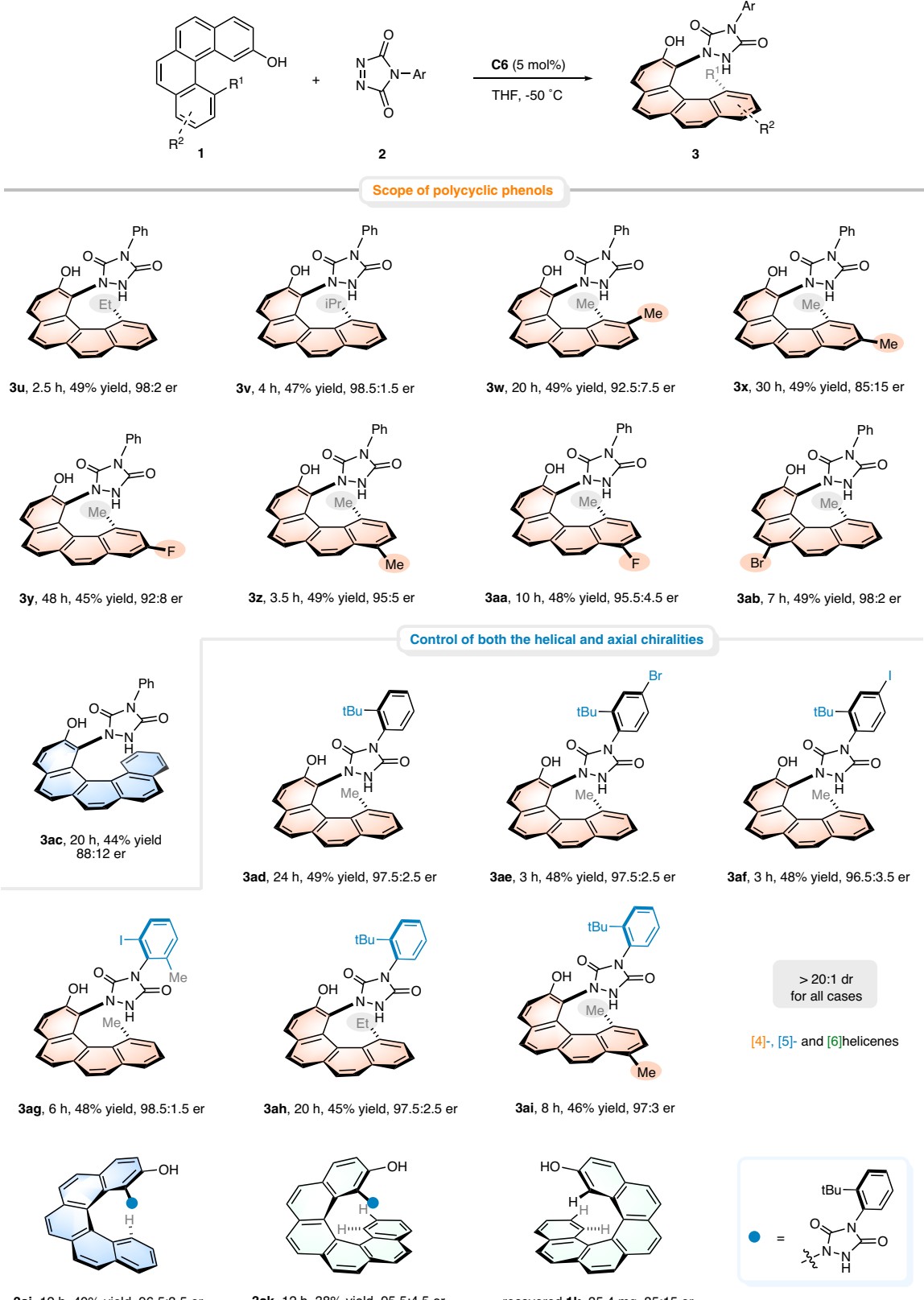

**Fig. 4 | Substrate scope with respect to polycyclic phenols and scope of the synthesis of helicenes bearing a stereogenic C-N axis.** Reaction conditions: 0.2 mmol **1** (1.0 equiv), 0.1 mmol **2** (0.5 equiv), and 5 mol% **C6** in THF (1.0 mL) at −50 °C. Isolated yields based on **1** are shown. The er values were determined by HPLC analysis using a chiral stationary phase.

relatively bulky isopropyl groups were all well tolerated, and the corresponding amination products were obtained in excellent yields and enantioselectivities (**3u**, **3v**). Substitutions at the 10 and 11 positions of

the terminal aromatic ring with a fluorine atom or a methyl group saw a slight decrement in enantioselectivities (**3w-3y**, 85:15-92.5:7.5 er) with increased steric hindrance effect, but led to consistently excellent

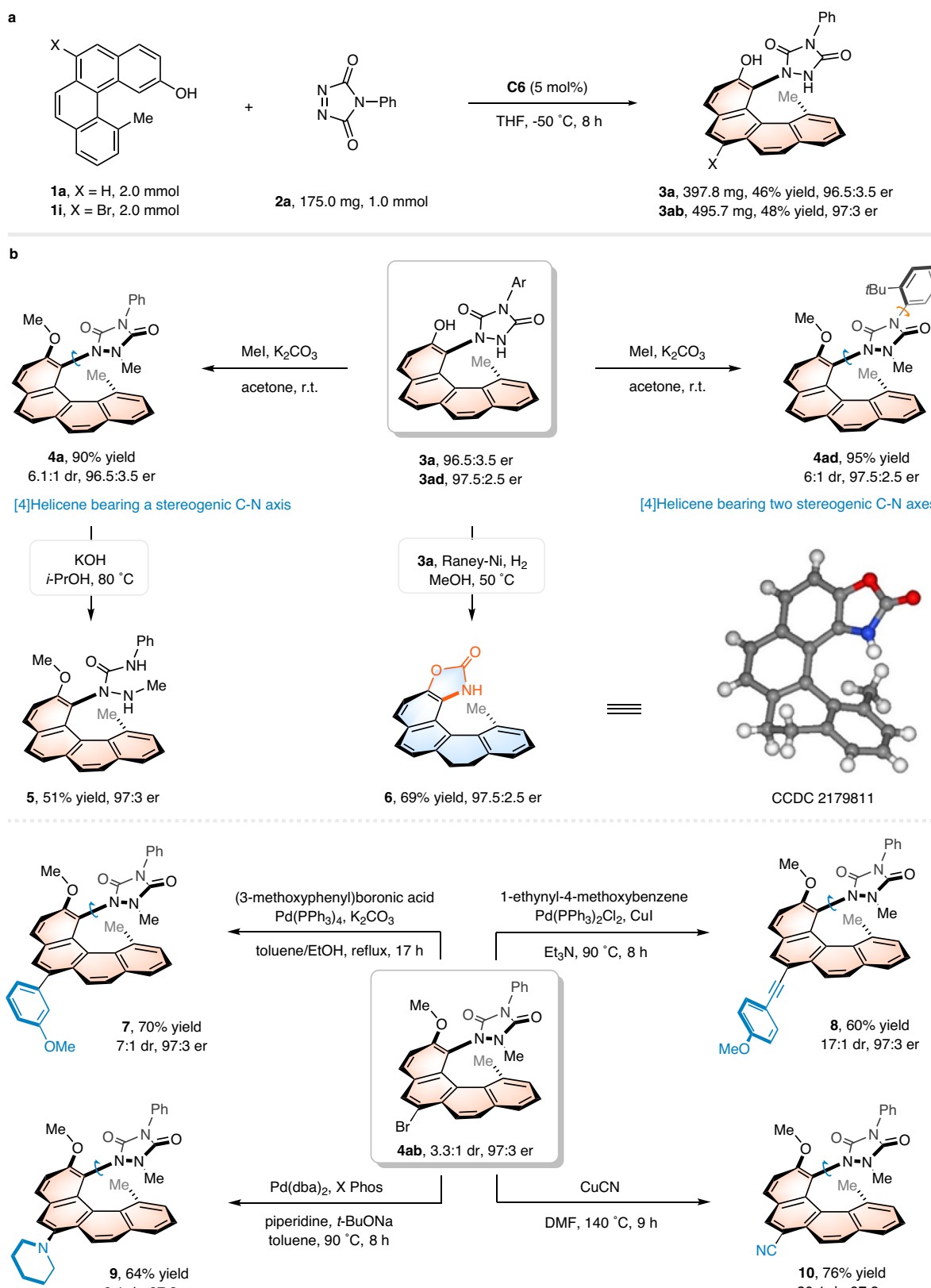

**Fig. 5 | Reaction scale-up and products transformations. a** Large-scale preparation of **3a** and **3ab**. **b** Synthetic transformations of **3a**, **3ad** and **4ab**. Isolated yields based on the substrate are shown.

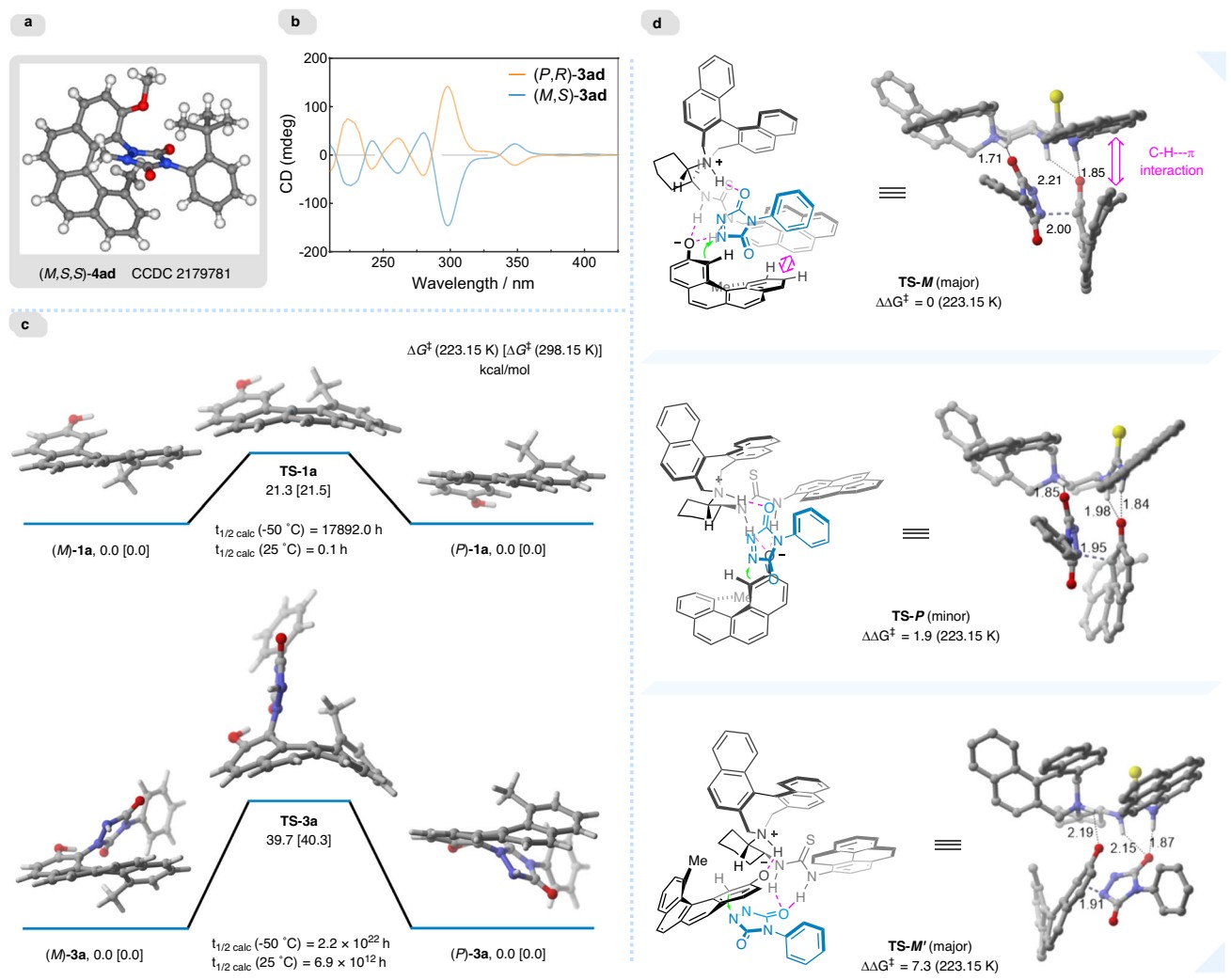

**Fig. 6 | Mechanistic studies. a** X-ray structure of (*M,S,S*)-**4ad**. **b** CD spectra of (*M,S*)-**3ad** (1.0 × 10⁻³ M) and its enantiomer (*P,R*)-**3ad** (1.0 × 10⁻³ M) in ethanol at room temperature. **c** Computed enantiomerization of **1a** and **3a**. **d** DFT calculated nucleophilic addition transition states with their relative free energies (in kcal/mol) and geometrical parameters (in Å).

chemical yields (45-49% yield). Such a decline in enantioselectivity might be attributed to the steric repulsion between the substituent at the terminal ring of the polycyclic phenols and the pyrenyl group of catalyst. In addition, halogen or methyl group could also be installed at the C-6 or C-9 position of substrate **1**, which appeared to be inconsequential to the reaction outcomes, delivering the desired [4]helicenes with commendable yields and enantioselectivities (**3z**-**3ab**, up to 49% yield and 98:2 er). To further extend the application of the present method to the enantioselective synthesis of [5]helicenes, racemic 2-[5] helicenol **1j** was recruited to undergo the current protocol. We were gratified to observe the smooth formation of **3ac** in 44% yield, albeit with relatively lower er value (88:12 er). It should be noted that the recovered **1j** was identified to be with an er value of 79.5:20.5, suggesting a reaction profile of catalytic kinetic resolution of configurationally unstable helical substrates at low temperature (−50 °C).

Encouraged by the above success, we then wondered whether this approach could be utilized to construct stereochemically complex molecules, especially those with multiple stereogenic elements. To the best of our knowledge, simultaneous control of helical and other chiralities via a catalytic enantioselective process represents a formidable challenge and has rarely been reported[56,57]. Towards this goal, the reactions of diazodicarboxamides with a sterically bulky *t*-butyl group at the *ortho*-position of phenyl ring were further performed

(Fig. 4), and surprisingly, supplied the respective helically chiral products bearing a stereogenic C-N axis in 45-49% yields with excellent stereoselectivities (**3ad**-**3af**, **3ah** and **3ai**, 96.5:3.5-97.5:2.5 er, >20:1 dr). Remarkably, stereochemically complex [5]- and [6]-helicenes were also achievable with comparably high er values (**3aj** and **3ak**). **3ag** bearing a 2-iodo-6-methylphenyl ring that is found frequently in C-N axially chiral atropisomers failed to form a configurationally stable C-N axis, mainly because the intramolecular steric repulsion caused by the terminal ring of [4]helicene facilitates the axial rotation. The capacity of realizing remote axial enantiocontrol while controlling helical sense selectivity again confirmed the broad generality of this catalytic enantioselective approach to derive structurally diverse helically chiral [4]-, [5]- and [6]helicenes.

### Synthetic applications

To demonstrate the scalability and practicality of this protocol in the synthesis of helically chiral chemicals, 2.0 mmol scale preparations of [4]helicenes **3a** and **3ab** under standard conditions were conducted (Fig. 5a). To our delight, the reactions proceeded smoothly and there was no obvious erosion of yield and enantioselectivity to be observed, suggesting a potential for large-scale chemical production of this method. Furthermore, some representative transformations of the helically chiral products **3a** and **3ad** were exhibited to further validate

the synthetic value of this methodology (Fig. 5b). For example, double methylation of **3a** with iodomethane was easily realized in the presence of K₂CO₃ at room temperature, by which a configurationally stable C-N axis was established in a diastereoselective fashion, giving [4]helicene **4a** in high yield with acceptable stereoselectivity (90% yield, 6.1:1 dr, 96.5:3.5 er). The configurationally stability of C-N axes in **4a** over **3a** was supported by DFT calculations. C-N bond rotation in **3a** shows a medium barrier of 15.4 kcal/mol, while bond rotation in **4a** is kinetically unfavorable with a higher barrier of 29.5 kcal/mol, resulting from the steric hindrance of the methyl substituents. Similarly, [4] helicene **4ad** bearing two stereogenic C-N axes could also be accessed. Subjecting **3a** to Raney Ni catalyst under a hydrogen atmosphere, a N−N bond cleavage and subsequent intramolecular cyclization occurred to generate biaryl compound **6** in 69% yield with excellent stereoretention. Notably, the urazole ring could also be cleaved by treating **4a** with KOH in isopropanol to deliver [4]helicene **5** in 51% yield. Additionally, diverse modifications of the main helical skeleton of **4ab** were also exhibited. Substituents such as aryl, alkynyl, and secondary amine groups can be readily introduced at the 6-position of [4]helicene **4ab** by involving Pd-catalyzed Suzuki-Miyaura, Sonogashira and Buchwald-Hartwig amination reactions, respectively. Treating **4ab** with CuCN at 140 °C produced the corresponding cyanated [4] helicene **10** with a yield of 76%.

## Mechanistic studies

The helical structure of **4ad** was unambiguously demonstrated by its X-ray crystallographic analysis (Fig. 6a), and the corresponding absolute configuration was definitely assigned to be (*M,aS,aS*) on the basis of the Flack parameter (0.07(13)). The configurations of other newly synthesized helicenes were inferred accordingly. In addition, the circular dichroism spectra of (*P,aR*)-**3ad** and (*M,aS*)-**3ad** were recorded to investigate the chiroptical properties, which display mirror images and clear Cotton effects at around 226, 242, 261, 280, 298, and 348 nm, respectively (Fig. 6b). Then, density functional theory calculations were conducted using the M11-L functional to obtain the interconversion profiles of the two helical enantiomers of compounds **1a** and **3a**, respectively (Fig. 6c). The relatively lower calculated barrier to enantiomerization for [4]helicene **1a** (ΔG‡ = 21.5 kcal/mol) revealed its configurational lability, corresponding to a half-life of 0.1 h at room temperature. In comparison, the barrier to enantiomerization for 1,12-disubstituted [4]helicene **3a** is significantly high (~40 kcal/mol), arguing for the high configurational stability of the *M/P*-configured **3a**.

Furthermore, based on the control experiments (Supplementary Fig. 5) and absolute configuration of amination products, we computationally explored the nucleophilic addition transition states of **1a** to **2a** to gain insight into the origin of the stereochemistry of this catalytic kinetic resolution process (Fig. 6d). The calculations revealed that catalyst **C6** brings the two reactants into close proximity via hydrogen bonding interactions and locks the relative orientation of the two substrates through their non-covalent interactions with the naphthalene skeleton and the pyrenyl substituent in the catalyst. In the most favorable transition state **TS-*M***, the bifunctional catalyst utilizes the thiourea N-H group to stabilize the anionic nucleophile (*M*)-**1a**, while the alkylammonium ion acts as a Brønsted acid to activate the diazodicarboxamide **2a**. The reaction with (*P*)-**1a** is relatively disfavored by 1.9 kcal/mol in the current catalytic system, largely due to the absence of C-H···π interactions between **1a** and the pyrenyl substituent of catalyst **C6** (Fig. 6d, **TS-*P***). Additionally, the traditional activation

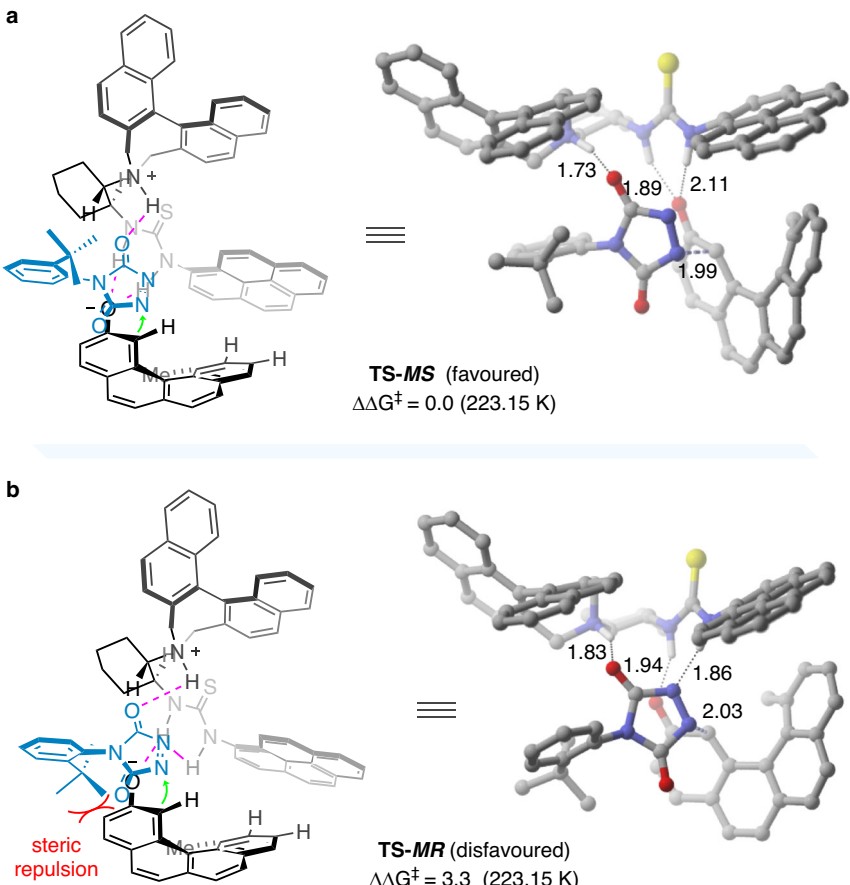

**Fig. 7 | Computational investigations on the origin of C-N axial chirality induction (energy in kcal/mol and bond lengths in Å). a** C-N bond forming transition state leading to the (*M,S*)-configured product. **b** C-N bond forming transition state leading to the (*M,R*)-configured product.

model with the tight alkylammonium-anionic **1a** ion pair is unlikely to be involved, owing to the energy-demanding transition state **TS-*M'***.

With the favored activation model established, we next intended to unravel the origin of C-N axial chirality induction. The optimized C-N forming TSs that give rise to the major and minor diastereoisomers of the product **3ad** are shown in Fig. 7. In the transition states **TS-*MS*** and **TS-*MR***, the phenyl ring of the diazodicarboxamide compound **2u** tends to be oriented away from the terminal ring of the polycyclic phenolate, in contrast to the arrangement observed in **TS-*M***. Transition state **TS-*MS***, which leads to the major (*M,S*)-configured product, is more favored than **TS-*MR*** by 3.3 kcal/mol. This preference arises from the decreased steric repulsion between the *tert*-butyl group on **2u** and the polycyclic phenolate.

## Discussion

In conclusion, we successfully developed a terminal *peri*-functionalization strategy for the catalytic enantioselective preparation of helically chiral molecules. The highly enantioselective synthesis of [4]carbohelicenes has been accomplished via an organocatalyzed intermolecular electrophilic aromatic amination reaction of 2-hydroxybenzo[*c*]phenanthrene derivates with diazodicarboxamides. The capacity of simultaneous control of helical and remote C-N axial chiralities made the current protocol especially intriguing, allowing for the constructing of enantioenriched [4]-, [5]- and [6]helicenes featuring an urazole scaffold with both structural diversity and stereochemical complexity in good to excellent yields and enantioselectivities. The mechanism investigation suggests that excellent enantiocontrol stems from a catalytic kinetic resolution process of configurationally unstable polycyclic phenols under low temperature. The diverse synthesis of previously elusive optically pure helicenes based on the strategy reported herein is currently in progress.

## Methods

### General procedure for catalytic enantioselective synthesis of 1,12-disubstituted [4]helicenes

The mixture of polycyclic phenols **1** (0.2 mmol, 1.0 equiv) and catalyst **C6** (0.01 mmol, 0.05 equiv) was dissolved in 1.0 mL of THF and stirred at room temperature for 10 min. Then, the solution was cooled to −50 °C and stirred for another 10 min before diazodicarboxamides **2** (0.1 mmol, 0.5 equiv) was added. The resulting mixture was stirred at this temperature until the complete consumption of **2**. After monitored by TLC, the solvent was removed under reduced pressure, and the residue was purified by silica gel column chromatography to afford the desired products **3**.

## Data availability

Experimental procedures, characterization data, copies of NMR spectra and computational details that support the findings of this study are available within the main text and its supplementary information files. Coordinates of the optimized structures are provided in the source data file. All other data are also available from the corresponding authors upon request. The X-ray crystallographic coordinates for structures reported in this study have been deposited at the Cambridge Crystallographic Data Centre (CCDC) under deposition numbers 2179781 (**4ad**) and 2179811 (**6**). These data can be obtained free of charge from The Cambridge Crystallographic Data Centre via www.ccdc.cam.ac.uk/data_request/cif. Source data are provided in this paper.

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

## Acknowledgements

We are grateful for the financial support from the National Natural Science Foundation of China (22001103 and 22371099 for X.L.), CAMS Innovation Fund for Medical Sciences (2019-I2M-5-074 for R.W.), Program for Chang-Jiang Scholars and Innovative Research Team in University (IRT_15R27 for R.W.), and the Science-Technology Foundation for Young Scientist of Gansu Province (23JRRA1115 for X.L.).

## Author contributions

X.L. and R.W. conceived and directed the project; X.L., B.Z., H.Z., J.Z., A.C., and F.W. conducted all experimental work; X.Z. conducted the computational work; X.L. and X.Z. analyzed the data; X.L. and R.W. wrote the manuscript with proofreading from all authors.

## Competing interests

The authors declare no competing interests.
