## [Peer Review File · Nature Communications]

Enantioselective Synthesis of [4]Helicenes by Organocatalyzed Intermolecular C-H AminationReviewers' Comments:

Reviewer #1:

Remarks to the Author:

In this manuscript, the authors have achieved the highly enantioselective preparation of [4]carbohelicenes through an organocatalytic intermolecular amination reaction between 2-hydroxybenzo[c]phenanthrene derivatives and diazodicarboxamides a terminal peri-functionalization strategy. The unique feature of this protocol lies in its ability to simultaneously control the helical and remote C-N axial chiralities, which adds an intriguing aspect to the study. Utilizing this approach, the authors were able to construct enantiopure [4]- and [5]helicenes with an urazole scaffold, showcasing both structural diversity and stereochemical complexity. The yields and enantioselectivities achieved were consistently good to excellent. Mechanistic investigations suggest that the excellent enantiocontrol observed stems from a catalytic kinetic resolution process of configurationally unstable polycyclic phenols under low-temperature conditions. Furthermore, DFT calculations revealed that both hydrogen bonds and C-H...N interactions between the substrates and catalyst play critical roles in achieving the desired stereochemical control. The authors have done an excellent job in presenting their manuscript, providing a clear context to illustrate the importance of their work, and adequately discussing their findings. This reviewer recommends publication of the manuscript in Nature Communications after addressing following issues.

1. Regarding the substrate scope illustrated in Figures 3 and 4, the authors have extensively investigated various diazodicarboxamides. However, it remains unclear how the reaction would proceed if azodicarboxylates were introduced as substrates. I am very curious about the results.
2. What would be the potential for utilizing 2-aminobenzo[c]phenanthrene derivatives as substrates in this transformation?
3. There appear to be several notable issues in the Supplementary Information, particularly with regards to the messy NMR spectra. The authors' conversion of compounds 3 into 4 through methylation has introduced a new C-N axis and led to suboptimal diastereoselectivity, further complicating the NMR analysis. This complication could potentially be resolved by employing high-temperature NMR techniques along with the appropriate deuterated reagent. Once the NMR issue is resolved, it is expected that the HPLC traces will exhibit improved clarity and cleanliness.

Reviewer #2:

Remarks to the Author:

This manuscript describes results on the catalytic asymmetric synthesis of substituted [4]helicenes. As mentioned by the authors, catalytic asymmetric synthesis of helical chirality is undoubtedly a challenging task with limited examples. In this study, the authors have successfully achieved catalytic asymmetric synthesis of substituted [4]helicenes with helical chirality with high stereoselectivity by functionalizing the generally sterically intricate fjord region. It is interesting to note that they have also succeeded in simultaneously inducing C-N axial chirality as well as helical chirality. Furthermore, as for the origin of the chirality, DFT calculations reveal that both hydrogen bonding between substrate and catalyst and C-H...N interactions contribute to the ideal stereochemical control. The results are excellent, but the following points are questionable and require more detailed description and discussion for publication.

- (1) The helical chirality of the present reaction is based on the suppression of racemization between the P- and M-bodies by the conversion of the peri-position of the substrate to a larger substituent, resulting in the development of stable helical chirality. On the other hand, this reaction is carried out at -50 degrees. As a result, 1a has a sufficiently large racemization barrier at -50 degrees ($t_{1/2} = 35776$ h). Therefore, dynamic kinetic resolution is probably not possible because racemization of the substrate is less likely to occur. This suggests that the reaction is proceeding with kinetic resolution, and as a result, yields cannot exceed 50% at best. However, I find it contradictory that yields exceed 50% in all systems and that the products are obtained with high stereoselectivity. How can a rational explanation be given for this phenomenon?

(2) I would also like to see a more detailed explanation of the mechanism by which C-N axis chirality is induced. By selectively functionalizing one of the two nitrogen atoms in enantioposition, asymmetric desymmetrization occurs and axial chirality is induced. Please describe which enantioposition is selected to produce the corresponding axial asymmetry after taking a conformation that avoids intramolecular steric repulsion.

(3) The presence of the Me substituent at the 10-position in 3x is clearly less ee than in others (e.g. 3W and 3Z). What effect does the presence of the substituent at 10-position have on the transition state that determines the stereochemistry?

(4) What is the estimated activation energy of the C-N axial chirality in 4a?

(5) A helicene is a compound in which all rings are connected by pi-conjugation. Therefore, it is strange to call "five-ring-fused helicene" a molecule whose pi-conjugation is broken by reduction, as in 6. 6 is a compound that should rather be called a biaryl.

Reviewer #3:

Remarks to the Author:

The submitted manuscript describes the highly enantioselective synthesis of a series of conformationally locked helicene derivatives. The authors used an optimized chiral thiourea organocatalyst for the enantiocontrolled addition of enolates generated from benzo[c]phenanthrene-2-ol derivatives over various diazodicarboxamides (the authors call it formally "C-H amination"). The manuscript is well and clearly written and perfectly supported by XRD analyses. The work appears at first glance to be a major step forward in the field of catalytic enantioselective synthesis of helicenes, but it deserves a substantial criticism:

(1) The presented synthetic approach to helicenes has a very limited substrate scope, since only [4]helicene derivatives and one [5]helicene derivative have been synthesized. The scope is artificially inflated and makes only negligible chemical sense. What is the actual chemical value of the 3a-3t derivatives, which differ only by a distant substitution on the phenyl group of the diazodicarboxamides (Fig. 3a)? The same holds for the 3u-3ai derivatives (Fig. 4). The study focuses on the enantioselective synthesis of helicenes, but the authors do not go beyond [5]helicene derivatives, do not present any higher homologues or heteroanalogues, and do not introduce any additional useful functional groups. The synthetic elaboration of 3a/3ad is nice, but again very limited (Fig. 5).

(2) All organocatalytic reactions are performed at -50 °C with the reaction period of 1.5-48 h ("... until the complete consumption of starting material 2 ..." (see Methods). The authors use the racemic benzo[c]phenanthrene-2-ol derivatives 1 with half-life $t_{1/2}$ of 35776.5 h at -50 °C (ca 4 years; Fig. 6c). If racemization of 1 is virtually frozen at -50 °C and helicity 3 is virtually locked after the reaction (the racemization barrier increases from 21.3 to 39.7 kcal/mol; Figure 6c), then how is it possible that only one highly enantioenriched product 3 is eventually formed in mostly >80% yield? Clearly, deracemization of 1 is observed at a temperature at which it cannot take place. Is the intermediary enolate derived from 1 much less configurationally stable? The authors must clarify this controversy or the manuscript cannot be published. This fundamental problem of the study is even more apparent in the case of the product 3ac, which is formed in 88% yield and in 88:12 er from the configurationally much more stable 2-[5]helicenol 1j! Such values cannot be observed if only kinetic resolution is operating.

(3) The energy profile of racemization of 1a and 3a was calculated and the half-life $t_{1/2}$ was determined. However, for a racemization barrier of 21.3 kcal/mol, the half-life at -50 °C should be approximately 15800 h and not 35776.5 h (Fig. 6c). The authors seem to have described racemization as first-order kinetics and not as reversible first-order kinetics! The authors should also check other kinetic data.

(4) The authors state that "... and the perfect conservation of enantiopurity even after prolonged time at 80 °C accounted for a very high energy barrier to enantiomerization of this class of helicenes." Such a simplistic statement is not sufficient. The authors should experimentally measure the relevant racemization barrier assuming that the compound does not decompose at elevated temperature.

(5) The authors should use the term enantiopurity with caution as it is reserved for optical purity of

>99.999% (below the detection limit of the second enantiomer).

(6) The authors determined the absolute configuration of 4ad by XRD analysis. Based on this and the correlation of ECD spectra, the absolute configuration of at least some of the products 3 should be made.

(7) The authors praise their method of asymmetric helicene synthesis at the expense of previously published and cited methods. However, the usefulness and generality of their method is still very questionable. The introduction should be modified accordingly, including a fair and objective comparison of alternative approaches. As it is, it comes across as very biased and partial.

Responds to the reviewers' comments:

Reviewer #1 (Remarks to the Author)

Comment 1: Regarding the substrate scope illustrated in Figures 3 and 4, the authors have extensively investigated various diazodicarboxamides. However, it remains unclear how the reaction would proceed if azodicarboxylates were introduced as substrates. I am very curious about the results.

Response: We thank the reviewer for this valuable suggestion. Following the reviewer's comments, the reaction between polycyclic phenol **1a** and azodicarboxylate **2'** was performed under the standard catalytic system. Unfortunately, even after stirring at room temperature for 24 hours, no amination product was detectable. This result has been documented in the supplementary information as Supplementary Fig. 4a.

Comment 2: What would be the potential for utilizing 2-aminobenzo[*c*]phenanthrene derivatives as substrates in this transformation?

Response: We thank the reviewer for this insightful suggestion. As suggested, 2-aminobenzo[*c*]phenanthrene derivative **1a'** was synthesized and recruited to undergo the current protocol. However, the corresponding amination product **3a'** was obtained with only 2% ee. This result has been documented in the supplementary information as Supplementary Fig. 4b.

Comment 3: There appear to be several notable issues in the Supplementary Information, particularly with regards to the messy NMR spectra. The authors' conversion of compounds **3** into **4** through methylation has introduced a new C-N axis and led to suboptimal diastereoselectivity, further complicating the NMR analysis. This complication could potentially be resolved by employing high-temperature NMR techniques along with the appropriate deuterated reagent. Once the NMR issue is resolved, it is expected that the HPLC traces will exhibit improved clarity and cleanliness.

Response: We thank the reviewer for this kind suggestion. According to the reviewer's suggestion, ^1H NMR spectra of **3a** were collected in d_6 -DMSO at varying temperatures. It is worth noting that the effect of atropisomerism on the ^1H NMR spectra persisted, even at elevated temperatures of 140 °C. In this case, we performed an indirect characterization of the NMR spectra by derivatizing the

products. Besides, the HPLC traces for compounds **3a-3ak** are available in the supplementary information.

Special thanks to reviewer #1 for the good comments and constructive feedback.

Reviewer #2 (Remarks to the Author)

Comment 1: The helical chirality of the present reaction is based on the suppression of racemization between the *P*- and *M*-bodies by the conversion of the *peri*-position of the substrate to a larger substituent, resulting in the development of stable helical chirality. On the other hand, this reaction is carried out at -50 degrees. As a result, **1a** has a sufficiently large racemization barrier at -50 degrees ($t_{1/2} = 35776$ h). Therefore, dynamic kinetic resolution is probably not possible because racemization of the substrate is less likely to occur. This suggests that the reaction is proceeding with kinetic resolution, and as a result, yields cannot exceed 50% at best. However, I find it contradictory that yields exceed 50% in all systems and that the products are obtained with high stereoselectivity. How can a rational explanation be given for this phenomenon?

Response: We thank the reviewer for this comment. The reaction reported herein is indeed proceeding with kinetic resolution. It should be noted that the feeding ratio of polycyclic phenols **1** to diazodicarboxamides **2** was 2:1, with the amount of diazodicarboxamides **2** being set at a stoichiometric equivalent. During the experiment, we found that diazodicarboxamides **2** could be completely consumed. As mentioned in the footnotes of the figures in the main text, the yields we reported in this article were calculated based on diazodicarboxamides **2**; hence, most yields

exceeded 50%. If the yields were calculated based on polycyclic phenols **1**, the product yields would theoretically be less than 50%, as the reviewer mentioned. This discrepancy in the results is due to the difference in the calculation standards. For example, You's work adopted a yield calculation approach same to the one we used (*Nat. Chem.* 2020 12, 838–844).

Comment 2: I would also like to see a more detailed explanation of the mechanism by which C-N axis chirality is induced. By selectively functionalizing one of the two nitrogen atoms in enantioposition, asymmetric desymmetrization occurs and axial chirality is induced. Please describe which enantioposition is selected to produce the corresponding axial asymmetry after taking a conformation that avoids intramolecular steric repulsion.

Response: We thank the reviewer for this valuable suggestion. To gain insight into the origin of C-N axial chirality induction, the complexes between (*M*)-polycyclic phenolate, diazodicarboxamide **2u** and protonated catalyst **C6** were further optimized by DFT calculations. In the transition states **TS-MS** and **TS-MR**, the phenyl ring of the diazodicarboxamide compound **2u** tends to be oriented away from the terminal ring of the polycyclic phenolate. Transition state **TS-MS**, which leads to the major (*M,S*)-configured product, is more favored than **TS-MR** by 3.3 kcal/mol. This preference arises from the decreased steric repulsion between the *tert*-butyl group on **2u** and the polycyclic phenolate. These results have been included in the manuscript and supplementary information.

Comment 3: The presence of the Me substituent at the 10-position in **3x** is clearly less ee than in others (e.g. **3w** and **3z**). What effect does the presence of the substituent at 10-position have on the transition state that determines the stereochemistry?

Response: We thank the reviewer for bringing this problem to our attention. The DFT calculations revealed that besides hydrogen bonds, the C-H--- π interactions between polycyclic phenolate and catalyst **C6** also contributes to the ideal stereochemical control. Substituting at the 10- or 11-position of 12-methylbenzo[*c*]phenanthren-2-ol leads to significant steric hindrance between the terminal ring of the polycyclic phenolate and the pyrenyl group on catalyst **C6**. This increased steric repulsion diminishes the C-H--- π interactions and results in a lower enantiomeric ratio (*er*). A corresponding description “Such a decline in enantioselectivity might be attributed to the steric repulsion between the substituent at the terminal ring of the polycyclic phenols and the pyrenyl group of catalyst” was added in the manuscript.

Comment 4: What is the estimated activation energy of the C-N axial chirality in **4a**?

Response: DFT calculations reveal the rotational barrier of the axial C-N bond in **4a** to be 29.8 kcal/mol. For further details, refer to reference 72 in the manuscript or Fig. 6 in the supplementary information.

Comment 5: A helicene is a compound in which all rings are connected by pi-conjugation. Therefore, it is strange to call "five-ring-fused helicene" a molecule whose pi-conjugation is broken by reduction, as in **6**. **6** is a compound that should rather be called a biaryl.

Response: We thank the reviewer for this professional suggestion. The inaccurate description about compound **6** has been corrected.

We are grateful to Reviewer #2 once more for the positive comments and professional suggestions.

Reviewer #3 (Remarks to the Author)

Comment 1: The presented synthetic approach to helicenes has a very limited substrate scope, since only [4]helicene derivatives and one [5]helicene derivative have been synthesized. The scope is artificially inflated and makes only negligible chemical sense. What is the actual chemical value of the **3a-3t** derivatives, which differ only by a distant substitution on the phenyl group of the diazodicarboxamides (Fig. 3a)? The same holds for the **3u-3ai** derivatives (Fig. 4). The study focuses on the enantioselective synthesis of helicenes, but the authors do not go beyond [5]helicene derivatives, do not present any higher homologues or heteroanalogues, and do not introduce any additional useful functional groups. The synthetic elaboration of **3a/3ad** is nice, but again very limited (Fig. 5).

Response: We thank the reviewer for this comment. In drug development, modifying substituents at different locations on a molecule often leads to significant alterations in its biological function. From an organic chemistry perspective, variations in steric hindrance, electronic effects, and the placement of substituents frequently exert a substantial influence on the results of chemical reactions, particularly impacting enantioselectivity. For instance, in this study, using diazodicarboxamide **2u** with a tert-butyl substituent led to a [5]helicene product **3aj** with an enantiomeric excess (*ee*) of 93%. Conversely, employing the unsubstituted diazodicarboxamide **2a** as a substrate resulted in the corresponding [5]helicene **3ac** with a lower *ee* value of 76%. This is precisely why we explored the types and range of substituents on both the substrates. Similar approaches to investigate substrate scope have also been embraced by other literatures (*J. Am. Chem. Soc.* 2023, 145, 15708–15713; *Nature*, 2023, 621, 753–759; *J. Am. Chem. Soc.* 2021, 143, 12924–12929; see also references 33-61 in the manuscript).

Besides, we conducted the catalytic kinetic resolution of a [6]helicene **1k** employing the method described and were able to isolate the desired product **3ak** with an enantiomeric excess (*ee*) of 91%. This outcome showcases the potential of our methodology for the enantioselective synthesis of higher homologues or heteroanalogue. Furthermore, diverse modifications of the main helical skeleton of **4ab** were also exhibited. Substituents such as aryl, alkynyl and secondary amine groups can be readily introduced at the 6-position of [4]helicene **4ab** by involving Pd-catalyzed Suzuki-Miyaura, Sonogashira and Buchwald-Hartwig amination reactions, respectively. Treating **4ab** with CuCN at 140 °C produced the corresponding cyanated [4]helicene **10** with a yield of 76%. These results have been included in the manuscript.

Comment 2: All organocatalytic reactions are performed at $-50\text{ }^\circ\text{C}$ with the reaction period of 1.5–48 h (“... until the complete consumption of starting material **2** ...” (see Methods). The authors use the racemic benzo[*c*]phenanthrene-2-ol derivatives **1** with half-life $t_{1/2}$ of 35776.5 h at $-50\text{ }^\circ\text{C}$ (ca 4 years; Fig. 6c). If racemization of **1** is virtually frozen at $-50\text{ }^\circ\text{C}$ and helicity **3** is virtually locked after the reaction (the racemization barrier increases from 21.3 to 39.7 kcal/mol; Figure 6c), then how is it possible that only one highly enantioenriched product **3** is eventually formed in mostly >80% yield? Clearly, deracemization of **1** is observed at a temperature at which it cannot take place. Is the intermediary enolate derived from **1** much less configurationally stable? The authors must clarify this controversy or the manuscript cannot be published. This fundamental problem of the study is even more apparent in the case of the product **3ac**, which is formed in 88% yield and in 88:12 er from the configurationally much more stable 2-[5]helicenol **1j**! Such values cannot be observed if only kinetic resolution is operating.

Response: We thank the reviewer for bringing this problem to our attention. The reaction described herein does indeed proceed via kinetic resolution. In our work, the feeding ratio of polycyclic phenols **1** to diazodicarboxamides **2** was 2:1, with the amount of diazodicarboxamides **2** being set at a stoichiometric equivalent. In the reaction system, there are equal molar amounts of *P*-configured and *M*-configured polycyclic phenols **1** present alongside diazodicarboxamides **2**. During the experiment, we found that diazodicarboxamides **2** selectively react with *M*-configured polycyclic phenols **1** and could be completely consumed under the current catalytic system. As mentioned in the footnotes of the figures in the main text, the yields we reported in this article were calculated based on diazodicarboxamides **2**; hence, most yields exceeded 50%. If the yields were calculated based on polycyclic phenols **1**, the product yields would theoretically be less than 50%, as the reviewer mentioned. This discrepancy in the results is due to the difference in the calculation standards. It should be noted that Your’s work adopted a yield calculation approach same to the one we used (*Nat. Chem.* 2020 12, 838–844).

Comment 3: The energy profile of racemization of **1a** and **3a** was calculated and the half-life $t_{1/2}$ was determined. However, for a racemization barrier of 21.3 kcal/mol, the half-life at -50 °C should be approximately 15800 h and not 35776.5 h (Fig. 6c). The authors seem to have described racemization as first-order kinetics and not as reversible first-order kinetics! The authors should also check other kinetic data.

Response: We thank the reviewer for this professional suggestion and carefulness. The half-life data for compounds **1a**, **1j**, and **3a** have been revised. Please see the revised manuscript and supplementary information.

Comment 4: The authors state that "... and the perfect conservation of enantiopurity even after prolonged time at 80 °C accounted for a very high energy barrier to enantiomerization of this class of helicenes." Such a simplistic statement is not sufficient. The authors should experimentally measure the relevant racemization barrier assuming that the compound does not decompose at elevated temperature.

Response: We thank the reviewer for this comment. We have previously attempted to determine the enantiomerization barrier of compound **5** (94% ee) in mesitylene. However, even after heating at 160 °C for 12 hours, there was no change in the enantiomeric excess (ee) value to be detected. Furthermore, we recently tried to measure it using DMSO as the solvent, there was still no change in the ee value after heating at 185 °C for 12 hours. This indicates that the configuration of compound **5** is indeed very stable. For details, see the previously submitted or the revised supplementary information. Besides, the description that "... and the perfect conservation of enantiopurity even after prolonged time at 80 °C accounted for a very high energy barrier to enantiomerization of this class of helicenes." has been removed from the manuscript.

Comment 5: The authors should use the term enantiopurity with caution as it is reserved for optical purity of >99.999% (below the detection limit of the second enantiomer).

Response: We thank for the reviewer's carefulness and preciseness. We checked both the manuscript and supplementary information. The related description has been corrected.

Comment 6: The authors determined the absolute configuration of **4ad** by XRD analysis. Based on this and the correlation of ECD spectra, the absolute configuration of at least some of the products **3** should be made.

Response: We thank the reviewer for this comment. According to the reviewer, the circular dichroism spectra of (M)-**3a**, (M)-**3r**, and (M)-**3ac** were further recorded. A significant agreement between the CD signatures of these helicenes and those of (M,S)-**3ad** was observed. These results have been documented in the supplementary information as Supplementary Fig. 10.

Comment 7: The authors praise their method of asymmetric helicene synthesis at the expense of previously published and cited methods. However, the usefulness and generality of their method is still very questionable. The introduction should be modified accordingly, including a fair and objective comparison of alternative approaches. As it is, it comes across as very biased and partial.

Response: We thank the reviewer for this comment. In the introduction section, we want to provide a comprehensive summary and overview of the current research status and existing synthetic methods relating to helical molecules. We did not intend to diminish the significance of previous work, which is why we utilized the term “pioneering and elegant works” to acknowledge the advancements made by previous researchers. If our wording caused any misunderstanding, we genuinely apologize for any confusion it may have caused. Besides, we have made some revisions to the introduction section to ensure a more accurate representation. Please see the revised manuscript.

We thank reviewer #3 once more for the meticulous comments and suggestions.

Reviewers' Comments:

Reviewer #1:

Remarks to the Author:

The authors have addressed most of points raised. I would like to recommend the publication of this manuscript in Nature Communications without any delay.

Reviewer #2:

Remarks to the Author:

Based on the comments of the three reviewers, I believe that the manuscript has been appropriately revised and contains good content with deeper insights. In particular, I am convinced that the addition of the DFT calculations for the asymmetric origins makes them more compelling and not just a list of results. Questions and areas for improvement are addressed in the text, and I believe that the manuscript has reached a level that could be published in Nat. Commun.

Reviewer #3:

Remarks to the Author:

The statement from the original review "The scope is artificially inflated and makes only negligible chemical sense" still holds true. The study has nothing to do with biological activity; the type of substituent in the para position has virtually negligible effect (see er oscillations between 95.5:4.5 and 98:2 for the ten compounds 3a-3j).

The calculation of the reaction yields relative to the reagent and not relative to the substrate, where the key kinetic resolution occurs, is fundamentally incorrect and confusing to the reader. The yields would be half of what is reported. The fact that this incorrect concept was used in another publication (Nat. Chem. 2020, 12, 838) does not legitimize it. Recalculating the yields would be a clean solution. If the authors persist in their position, which is conceptually wrong, then they should explicitly state in the table footnotes that the yields were calculated relative to the reagent.

The authors have dealt with the other comments very well.

Responds to the reviewers' comments:

Reviewer #1 (Remarks to the Author)

The authors have addressed most of points raised. I would like to recommend the publication of this manuscript in Nature Communications without any delay.

Response: We thank the reviewer for the kind comments.

Reviewer #2 (Remarks to the Author)

Based on the comments of the three reviewers, I believe that the manuscript has been appropriately revised and contains good content with deeper insights. In particular, I am convinced that the addition of the DFT calculations for the asymmetric origins makes them more compelling and not just a list of results. Questions and areas for improvement are addressed in the text, and I believe that the manuscript has reached a level that could be published in Nat. Commun.

Response: We thank the reviewer for the kind comments.

Reviewer #3 (Remarks to the Author)

The statement from the original review "The scope is artificially inflated and makes only negligible chemical sense" still holds true. The study has nothing to do with biological activity; the type of substituent in the para position has virtually negligible effect (see er oscillations between 95.5:4.5 and 98:2 for the ten compounds 3a-3j).

The calculation of the reaction yields relative to the reagent and not relative to the substrate, where the key kinetic resolution occurs, is fundamentally incorrect and confusing to the reader. The yields would be half of what is reported. The fact that this incorrect concept was used in another publication (Nat. Chem. 2020, 12, 838) does not legitimize it. Recalculating the yields would be a clean solution. If the authors persist in their position, which is conceptually wrong, then they should explicitly state in the table footnotes that the yields were calculated relative to the reagent.

The authors have dealt with the other comments very well.

Response: Thanks for the kind suggestions. The yields reported in the manuscript and Supplementary Information file have been recalculated.